# QTL and Candidate Genes for Seed Tocopherol Content in ‘Forrest’ by ‘Williams 82’ Recombinant Inbred Line (RIL) Population of Soybean

**DOI:** 10.3390/plants11091258

**Published:** 2022-05-06

**Authors:** Dounya Knizia, Jiazheng Yuan, Naoufal Lakhssassi, Abdelhalim El Baze, Mallory Cullen, Tri Vuong, Hamid Mazouz, Henry T. Nguyen, My Abdelmajid Kassem, Khalid Meksem

**Affiliations:** 1School of Agricultural Sciences, Southern Illinois University, Carbondale, IL 62901, USA; dounya.knizia@siu.edu (D.K.); naoufal.lakhssassi@siu.edu (N.L.); abdelhalim.elbaze@siu.edu (A.E.B.); cullenmallory@gmail.com (M.C.); 2Laboratoire de Biotechnologies & Valorisation des Bio-Ressources (BioVar), Department de Biologie, Faculté des Sciences, Université Moulay Ismail, Meknes 50000, Morocco; h.mazouz@fs-umi.ac.ma; 3Plant Genomics and Biotechnology Laboratory, Department of Biological and Forensic Sciences, Fayetteville State University, Fayetteville, NC 28301, USA; jyuan@uncfsu.edu (J.Y.); mkassem@uncfsu.edu (M.A.K.); 4Division of Plant Science and Technology, University of Missouri, Columbia, MO 65211, USA; vuongt@missouri.edu (T.V.); nguyenhenry@missouri.edu (H.T.N.)

**Keywords:** soybean, RIL, forrest, Williams 82, linkage map, tocopherol, SNP

## Abstract

Soybean seeds are rich in secondary metabolites which are beneficial for human health, including tocopherols. Tocopherols play an important role in human and animal nutrition thanks to their antioxidant activity. In this study, the ‘Forrest’ by ‘Williams 82’ (F×W82) recombinant inbred line (RIL) population (*n* = 306) was used to map quantitative trait loci (QTL) for seed α-tocopherol, β-tocopherol, δ -tocopherol, γ-tocopherol, and total tocopherol contents in Carbondale, IL over two years. Also, the identification of the candidate genes involved in soybean tocopherols biosynthetic pathway was performed. A total of 32 QTL controlling various seed tocopherol contents have been identified and mapped on Chrs. 1, 2, 5, 6, 7, 8, 9, 10, 12, 13, 16, 17, and 20. One major and novel QTL was identified on Chr. 6 with an R^2^ of 27.8, 9.9, and 6.9 for δ-tocopherol, α-tocopherol, and total tocopherol content, respectively. Reverse BLAST analysis of the genes that were identified in *Arabidopsis* allowed the identification of 37 genes involved in soybean tocopherol pathway, among which 11 were located close to the identified QTLs. The tocopherol cyclase gene (*TC*) *Glyma.06G084100* is located close to the QTLs controlling δ-tocopherol (R^2^ = 27.8), α-tocopherol (R^2^ = 9.96), and total-tocopherol (R^2^ = 6.95). The geranylgeranyl diphosphate reductase (*GGDR*) *Glyma.05G026200* gene is located close to a QTL controlling total tocopherol content in soybean (R^2^ = 4.42). The two methylphytylbenzoquinol methyltransferase (*MPBQ-MT*) candidate genes *Glyma.02G002000* and *Glyma.02G143700* are located close to a QTL controlling δ-tocopherol content (R^2^ = 3.57). The two γ-tocopherol methyltransferase (*γ-TMT*) genes, *Glyma.12G014200* and *Glyma.12G014300*, are located close to QTLs controlling (γ+ß) tocopherol content (R^2^ = 8.86) and total tocopherol (R^2^ = 5.94). The identified tocopherol seed QTLs and candidate genes will be beneficial in breeding programs to develop soybean cultivars with high tocopherol contents.

## 1. Introduction

Tocopherols and tocotrienols collectively constitute the tocochromanols family, known as Vitamin E. Tocochromanols are fat-soluble phenolic compounds, synthesized by photosynthetic organisms. In soybean, vitamin E is present almost exclusively as tocopherols. Tocopherols exist in four isoforms, α-tocopherol, γ-tocopherol, β-tocopherol, and δ-tocopherol which differ from each other by the number and the location of the methyl groups. α-tocopherol possesses three methyl groups, followed by γ-tocopherol and β-tocopherol that have two methyl groups, and finally δ-tocopherol with only one methyl group [1].

Tocopherols have an important role in human and animal nutrition thanks to their vitamin E activity. However, from a nutritional perspective among the four tocopherol isoforms, α-tocopherol is the most important due to the high vitamin E activity [2]. It also has been reported to play a role in the prevention of cardiovascular diseases and cancer [3,4]. In the human body, α-tocopherol is preferentially accumulated due to its affinity with the liver α-tocopherol transfer protein (α-TTP), which enriches plasma with α-tocopherol [5].

Soybean (*Glycine max* Merr.) is not only one of the main sources of vegetable oil and animal feed worldwide, but also used for production of biofuel, aquaculture feed, and as source of protein for the human diet due to a high protein (40–42%) and oil contents (18–22%) [6], which make it an important crop worldwide.

Soybean seeds are rich in secondary metabolites beneficial for human health including tocopherols. Total tocopherol content is relatively high in soybean seeds compared to other oilseeds crops, and γ-tocopherol is the predominant form, while α-tocopherol content is less than 10% of the total tocopherol content [3,4,7].

Since soybean oil provides 30% of the total worldwide oil consumption and ~70% of the vitamin E in the American diet comes from soybean oil, developing soybean cultivars with high seed α-tocopherol contents could have tremendous positive effects on the health benefits associated with eating soybeans and their market value.

In soybean, tocopherol seed content and composition vary from one cultivar to another, being controlled by several genetic and environmental factors. These factor make it one of the most complex quantitative traits [8] and many studies have focused on investigating the genetic and molecular factors underlying this trait [9,10,11,12].

A ‘TK780’ by ‘B04009’ RIL population (*n* = 94) was used to identified six QTLs involved in α-tocopherol biosynthesis on Chr. 9, 11 and 12 [7]. Also, using a ‘OAC Bayfield’ × ‘OAC Shire’ RIL population across three locations over 2 years, and 151 SSR markers, 26 SSR markers linked to QTLs with individual and total tocopherol content across 17 chromosomes were identified [13]. Likewise, using a ‘Beifeng 9’ by ‘Freeborn’ RIL population in six environments, 18, 13, 11, and 13 QTLs associated with seed α-tocopherol, γ-tocopherol, δ-tocopherol, and total tocopherol contents were identified respectively [14].

Tocopherol biosynthesis takes place at the plastid’s envelope, where a combination of two precursors derived from different pathways occurs. The homogentisic acid (HGA), a product of the cytosolic shikimate pathway, is used to form the aromatic ring of tocopherols, while phytyl diphosphate (PDP), a product of either the methylerytrithol phosphate (MEP) pathway or the phytol recycling pathway [15], forms the prenyl tail. The condensation of these two precursors is catalyzed by the homogentisate phytyl transferase (HPT) and creates 2-methyl-6- phytyl-1,4-benzoquinol (MPBQ), which can be further methylated by MPBQ methyltransferase (MPBQ-MT) to 2,3-dimethyl-6-phytyl-1,4-benzoquinone (DMPBQ). The cyclization of the MPBQ and DMPBQ by the tocopherol cyclase produces γ-tocopherol and δ-tocopherol, respectively. The conversion of γ-tocopherol and δ-tocopherol to α-tocopherol and β-tocopherol is catalyzed by the γ-tocopherol methyltransferase (γ-TMT) and it represents the last step of the tocopherol biosynthesis pathway.

Many studies have elucidated the genes involved in tocopherol pathway in *Arabidopsis*. The tyrosine produced by the shikimate pathway is catalyzed by Tyrosine Aminotransferase (TAT) resulting in the formation of P-hydroxy Phenyl Pyruvate (HPP). The HPP will be catalyzed by p-hydroxyphenylpyruvate dioxygenase (HPPD) to produce the homogentisic acid, this enzyme is controlled by PDS1. In *A. thaliana,* mutants of *pds1* have shown a lack of tocopherols and plastoquinone with a lethal photobleached phenotype, this result showed the importance of PDS1 in the tocopherol biosynthesis pathway [16]. The overexpression of the *PDS1* gene in tobacco leaves or in *A. thaliana* seeds only gave moderately increased tocopherol concentrations [17,18]. The phytyl diphosphate (PDP) can be derived, either from the MEP pathway after reduction of geranylgeranyl diphosphate (GGDP) by the Geranylgeranyl Diphosphate Reductase (GGDR) enzyme, or from the phytol recycling pathway. Many studies have investigated the phytol recycling pathway and have shown that mutants of *vte5-1* are devoid of phytol kinase. Also, *vte5-1* mutants have shown a reduction in total tocopherol content in seeds and leaves with 80% and 65% respectively, compared to the wild type [19]. The *VTE5* gene controls the phytol kinase that catalyzes the phytol phosphorylation producing Phytolmonophosphate which is catalyzed by Phytolphosphate kinase VTE6 leading to phytyl diphosphate (PDP) formation. Arabidopsis *vte6* mutants have shown tocopherol deficiency in leaves and a reduction in plant growth and seed longevity. The overexpression of the *VTE6* gene resulted in a two-fold increase in PDP that resulted in higher γ-tocopherol accumulation in seeds [20]. Homogentisate phytyl transferase (HPT) catalyzes the condensation of HGA and PDP to produce 2-methyl-6-phytyl-1,4- benzoquinone (MPBQ). In *Arabidopsis*, the HPT enzyme is encoded by the *VTE2* gene [21,22].

The *Arabidopsis* devoid of VTE2 have shown a complete deficiency in all tocopherol derivatives and all pathway precursors, which means that this is a crucial step in the tocopherol biosynthetic pathway [23]. The MPBQ-MT enzyme is encoded by the *VTE3* gene, which is a limiting step in producing α- and γ-tocopherol. In *Arabidopsis*, *vte3-2* mutants were lacking in α- and γ-tocopherol and exhibited a pale green phenotype, abnormal chloroplasts and did not survive beyond the seedling stage [24,25].

The *VTE1* gene catalyzes the conversion of 2-methyl-6-phytyl-1,4- benzoquinone (MPBQ) and 2,3-dimethyl-5-phytyl-1,4-benzoquinone (DMPBQ) to δ-tocopherol and γ-tocopherol, respectively. *vte1* mutants have a nonfunctional tocopherol cyclase enzyme (TC) and are totally devoid of all tocopherol forms, and accumulate DMPBQ, the γ-tocopherol precursor [26]. The overexpression of the *VTE1* gene in *Arabidopsis* plants showed an improvement in total tocopherol by 7-fold in leaves, as well as a major shift from α- to γ-tocopherol [27].

The VTE4 gene encodes the γ-tocopherol methyltransferase (γ-TMT) that catalyzes the methylation of the γ-tocopherol and δ-tocopherol to produce α-tocopherol and β-tocopherol, respectively. The co-expression of both *At-VTE3* and *At-VTE4* in soybean showed an accumulation of >95% of α-tocopherol, in addition to a 5-fold increase of seed vitamin E activity [28].

In soybean, few genes have been reported to play a role in seed tocopherol content. These genes are *γ-TMT1, γ-TMT2,* and *γ-TMT3* mainly [29]. Also, the Di-glucose binding protein with Leucine-rich repeat domain gene (*Glyma.02G099800*); Eukaryotic aspartyl protease family protein gene (*Glyma.02G100800*); Cytochrome b561/ferric reductase transmembrane protein family gene (*Glyma.02G101300*); Transmembrane amino acid transporter family protein gene (*Glyma.02G098200*); and plant U-box 26 gene (*Glyma.02G102900*) were identified and determined to be significantly associated with α-tocopherol, γ-tocopherol, δ-tocopherol and total tocopherol in soybean seeds [30].

In this study, the genetic factors associated with tocopherol content in soybean were investigated, QTL for seed α-tocopherol, β-tocopherol, δ-tocopherol, γ-tocopherol, and total tocopherol contents were mapped, the link between the biosynthesis genes for tocopherol and soybean seed tocopherol content was studied, and the *in-Silico* tocopherol biosynthetic pathway in soybean was reconstructed.

## 2. Results

### 2.1. The SNP-Based Genetic Map

The SNP-Based genetic map used in this study was described previously and identified QTLs that control seed isoflavone contents [31]. The map which covered 4029.9 cM, was composed of 2075 SNP markers, and was based on 306 RILs of F × W82 [31].

### 2.2. Tocopherol Contents Frequency Distribution, Heritability, and Correlation

The frequency distributions among different tocopherol contents were not always normal in the FxW82 RIL population based on Shapiro–Wilk’s method for normality test. Only total-tocopherol 2017 (T-Toc-2017) and δ-tocopherol 2020 (δ-Toc-2020) were normally distributed. The positive or negative skewness and kurtosis value (>3) were also identified in the RIL population (Table 1; Figure 1).

Each tocopherol component also showed a different degree of variation in the parameters of traits, and the variability appeared to not be greatly impacted by different environments. α-tocopherol 2017 (α-Toc-2017), displayed the highest coefficient of variation (CV) value (83.64%); however, the CV of (α-Toc-2020) was 55.49% indicating that phenotypic variability among tocopherol contents was constant year over these 2 years.

The broad sense heritability (h^2^) of (µg/g of dry seed weight) for seed α-tocopherol (α-Toc), δ-tocopherol (δ-Toc), γ+β-tocopherol ((γ+β)-Toc), and total tocopherol (T-Toc) contents (in µg/g of dry seed weight) across two different years appeared to be quite diverse. δ-Toc had the highest heritability (71%) and the h^2^ for T-Toc was 41% (Table 2). However, the h^2^ values for (γ+β)-Toc and α-Toc were negative (*−*41% and *−*61%, respectively) implying that there was biologically meaningful phenotypic repulsion among these traits. The high heritability of seed δ-Toc contents suggested that a large portion of phenotypic variation could be detected in the mapped QTL. The RILs-Year interactions still played a significant role in the molecular formation among tocopherols in soybean seeds based on our two-way ANOVA analysis because the σGE^2^ is relatively high (data not shown). It should be used as a parameter for trait improvement.

Due to cost effect of this undergraduate student-centered project, only technical replicates could be applied, and F value and probability could not be generated from the dataset (Table 2). Hence, we only calculated the Sum Sq and Mean Sq to determine σG^2^ and σGE^2^ for each trait (Table 2) using type I sum of squares (ANOVA (model)) function in R program but not σe^2^ due to limited replicates.

### 2.3. Seed Tocopherol Contents QTL

We used both the interval mapping (IM) and composite interval mapping (CIM) methods of WinQTL Cartographer 2.5 [32] to identify QTLs that control seed α-Toc, δ-Toc, (γ+β)-Toc, and T-Toc contents; however, only QTLs identified by CIM method with LOD scores >2.5 are reported here. A total of 32 QTL that control these seed tocopherols contents have been identified and mapped on Chr. 1, 2, 5, 6, 7, 8, 9, 10, 12, 13, 16, 17, and 20 in this RIL population grown in both years (2017 and 2020) (Table 3, Appendix A).

Five QTLs controlling α-tocopherol content in soybean were identified on Chrs. 6, 1 and 2 (Table 3, Appendix A). The qα-Toc-2-IL-2017 (192.6–197.6 cM) and qα-Toc-3-IL-2020 (195–197 cM) were collocated on Chr. 06. Additionolly, eight QTLs underlying δ-tocopherol content were identified on Chrs. 1,2,6,8,16,20. The qδ-Toc-3-IL-2017 located on Chr.6 explains 27.9% of the phenotype (Table 3, Appendix A). For the γ+β tocopherol content, ten QTLs were identified on Chrs. 2,6,8,12,13, and 16 (Table 3, Appendix A). Twelve QTLs controlling total tocopherol content were identified and mapped on Chrs. 1, 5, 6, 7, 8, 9, 10, 12, 16, 17, 20 (Table 3, Appendix A).

### 2.4. In Silico Reconstruction of the Tocopherol Biosynthetic Pathway in Soybean

The tocopherol biosynthetic pathway has been investigated in the model plant Arabidopsis thaliana. The genes and compounds involved in that pathway were previously reported [33]. To reconstruct the tocopherol biosynthesis pathway in soybean, the reverse BLAST of these genes was conducted using SoyBase.

Thirty-seven candidate genes underlying the soybean’s tocopherol pathway were identified (Figure 2). In the Shikimat pathway five candidate genes were identified for Tyrosine Aminotransferase (TAT) including *Glyma.06g235500, Glyma.06g235900, Glyma.12g161500, Glyma.12g205900,* and *Glyma.13g295000*. Two candidate genes were identified for hydroxyphenylpyruvate dioxygenase (HPPD) (PDS1), *Glyma.14G030400,* and *Glyma.02G284600*. In the last step of the MEP pathway three candidate genes underlying the geranylgeranyl diphosphate reductase (GGDR) that catalyzes the production of phytyl diphosphate (PDP) from geranylgeranyl diphosphate (GGDP), were identified, *Glyma.02G273800, Glyma.05G026200,* and *Glyma.17G100700* (Figure 2).

For the phytol recycling pathway, one candidate gene was identified to be the phytol kinase (VTE5) *Glyma.20G190100*, and two candidate genes were identified to be the phytol-phosphate kinase (VTE6) *Glyma.13G265200*, and *Glyma.12G233800* (Figure 2).

In the core tocopherol pathway, twelve candidate genes were identified for the HPT (VTE2), *Glyma.17G061900*, *Glyma.13G097800*, *Glyma.03G033100, Glyma.10G070100, Glyma.01G134600, Glyma.10G295300, Glyma.20G245100, Glyma.08G274800, Glyma.10G070300, Glyma.13G152814, Glyma.13G152780,* and *Glyma.13G152746,* four were identified for the TC (VTE1), *Glyma.04G082500, Glyma.06G084100, Glyma.04G082300,* and *Glyma.04G082400*. Five candidate genes were identified for the MPBQ-MT (VTE3), *Glyma.02G143700, Glyma.10G030600, Glyma.02G002000, Glyma.20G211500,* and *Glyma.10G178600*, in addition to three for the γ-TMT (VTE4), *Glyma.09G222800, Glyma.12G014200,* and *Glyma.12G014300* (Figure 2).

### 2.5. The Association between the Identified Tocopherol Pathway Candidate Genes and the Identified Tocopherol QTL

Among the identified candidate genes, 11 were located close to the identified QTLs on Chrs. 2, 5, 6, 10, 12, and 17 (Table 4, Figure 2). These candidate genes include the tocopherol cyclase candidate (TC) gene *Glyma.06G084100* that is located close to seven seed tocopherol QTLs controlling δ-tocopherol, α-tocopherol, and total tocopherol on Chr. 6 (Table 4).

On Chr. 2, the MPBQ-MT candidate genes *Glyma.02G002000* and *Glyma.02G143700* are located close to qδ-Toc-2-(2017). The *Glyma.02G002000* candidate gene is also located close to qα-Toc-2-(2020) and q(γ+ß)-Toc-1-(2020) (Table 4, Figure 2). The γ-TMT candidate genes *Glyma.12G014200* and *Glyma.12G014300* are positioned near to QTLs controlling γ+ß tocopherol and total tocopherol (Table 4, Figure 2). The HPT candidate gene *Glyma.17G061900*, and the GGDR candidate gene *Glyma.17G100700* are located close to QTLs controlling total tocopherol on Chr. 17 (Table 4, Figure 2). *Glyma.05G026200* is a GGDR candidate gene that is positioned near to a QTL underlying total tocopherol on Chr. 5 (Table 4, Figure 2).

### 2.6. Association between the Identified Candidate Genes and the Previously Reported QTL

Mapping the identified genes to the previously reported QTL regions associated with soybean seeds tocopherols was done using data from SoyBase and previous studies describing the QTL underlying tocopherol contents in soybean [7,13,34,35]. Six candidate genes were located within the identified seed tocopherol QTLs and ten were very close to some of these regions (Table 5).

Among these QTLs, qαTC-9 QTL was collocated with the γ-TMT3 (*Glyma.09G222800*) [7]. Also, γ-TMT2 (*Glyma.12G014300*) and γ -TMT1 (*Glyma.12G014200*) are located 703 kb and 711 kb, respectively, apart from qαTC-12 QTL associated with α-tocopherol content [7]. Whereas the fourth tocopherol candidate gene (*Glyma.06G084100*) was located 9.7 Mbp from the QTVEC2_2 QTL underlying total seed tocopherol content identified earlier [34]. The HPT gene (*Glyma.17G061900*) is located 4 Mbp apart from qαγR-17 QTL associated with seed α-tocopherol content [7]. Moreover, *Glyma.04G082500, Glyma.04G082300,* and *Glyma.04G082400*, the tocophetol cyclase candidate genes, were located within qδTC-4 QTL associated with δ-tocopherol [7]. The MPBQ-MT genes *Glyma.02G002000* and *Glyma.02G143700* were located within two QTL controlling the seed γ-tocopherol content. The first one is the seed tocopherol, γ-1-5 QTL [34] (https://soybase.org/; accessed on 3 April 2022) and the second one is the seed tocopherol, γ-2-5 [35] (https://soybase.org/; accessed on 3 April 2022). The two HPT candidate genes *Glyma.10G070100* and *Glyma.10G070300* and the MPBQ-MT candidate gene *Glyma.10G030600* are located 3.3, 3.18, and 7.6 Mbp, respectively from the seed total tocopherol, T-Toc 2-2 QTL [35] (https://soybase.org/; accessed on 3 April 2022). The MPBQ-MT candidate gene *Glyma.20G211500* and the HPT candidate gene (*Glyma.20G245100*) were located 0.22 and 0.9 Mbp apart from the seed tocopherol, QTVEC2_2 QTL controlling total tocopherol content [35] (Table 5).

### 2.7. Organ-Specific Expression of the Identified Candidate Genes

To investigate the role of the identified 37 candidate genes, the expression analysis of these genes was performed in Williams 82 cv. using the publicly available RNA-seq database at SoyBase (https://soybase.org/; accessed on 3 April 2022). The tissues that were included in this dataset were leaves, nodules, roots, pods, and seeds. Amongst the 37 candidate genes, no RNAseq data was available for the TC candidate gene *Glyma.04G082400*, the HPT candidate gene *Glyma.08G274800*, and the MPBQ-MT candidate gene *Glyma.10G178600*. The rest of the tocopherol candidate genes presented different gene expression patterns. Most of the identified candidate genes were expressed in all the analyzed tissues except for the HTP candidate gene, *Glyma.03G033100*, that was not expressed in any of the tissues. While the two GGDR candidate genes, *Glyma.05G026200* and *Glyma.17G100700*, the HPT candidate gene, *Glyma.13G097800*, and the MPBQ-MT candidate gene, *Glyma.02G143700*, were highly expressed in flowers. The two GGDR candidate genes *Glyma.05G026200* and *Glyma.17G100700*, the MPBQ-MT candidate genes *Glyma.02G143700, Glyma.02G002000,* and *Glyma.10G030600*, the TC candidate gene *Glyma.06G084100*, and the γ-TMT candidate gene *Glyma.12G014300* were abundantly expressed in leaves. The GGDR candidate genes *Glyma.05G026200, Glyma.02G273800* and *Glyma.17G100700*, and the TAT candidate gene *Glyma.12G161500* were highly expressed in seeds. The TAT candidate genes *Glyma.06G235900* and *Glyma.12G205900*, the two GGDR candidate genes *Glyma.05G026200* and *Glyma.17G100700*, and the MPBQ-MT candidate genes *Glyma.02G143700* and *Glyma.10G030600* were highly expressed in pods. The γ-TMT candidate gene *Glyma.09G222800*, the GGDR candidate genes *Glyma.05G026200* and *Glyma.02G273800* were highly expressed in roots. Whereas the TAT candidate genes *Glyma.12G161500* and *Glyma.13G295000*, the GGDR candidate genes *Glyma.05G026200* and *Glyma.02G273800*, and the MPBQ-MT candidate gene *Glyma.10G030600* were highly expressed in the nodules (Figure 3A, Appendix A).

Amongst the identified candidate genes, eleven were located close to the tocopherol seed QTLs identified in FxW82 RIL population, in tocopherol seed content in soybean. *Glyma.05G026200* and *Glyma.17G100700* are highly expressed in seeds in Williams 82 cv., followed by *Glyma.02G002000*, *Glyma.02G143700, Glyma.10G030600, Glyma.12G014300,* and *Glyma.06G084100* that have moderate expression profiles in seeds. The rest of the genes have a low expression profile in seeds except for *Glyma.10G070100* and *Glyma.10G070300* that have a limited expression profile, with very low to no expression in seeds (Figure 3B).

## 3. Discussion

Tocopherols are lipophilic antioxidants that are important for human health due to their ability to prevent the oxidation of unsaturated fatty acids by scavenging the free radicals and prevent cell membrane damage [13]. Soybean seeds contain the highest tocopherol concentrations among all legume species [36]. The dominant tocopherol isoform in soybean seeds is γ-tocopherol with amounts reaching up to 70% of the total tocopherol content, while α-tocopherol isoform has a lower concentration of about 10% of the total tocopherol content. The α-tocopherol isoform has the highest vitamin E activity [4] and has the highest affinity with the hepatic tocopherol transfer protein. Therefore, improving soybean seed tocopherol composition and content is crucial.

Several studies have revealed the genetic and molecular bases underlying tocopherol content in soybean [7,13,14,29,30,34,35] as summarized recently in [37].

Among the QTLs identified in these studies, qαTC-9 QTL was collocated with the γ-*TMT3 Glyma.09G222800*; [7], however, the QTL identified in this study on Chr. 9 was more than 40 Mbp apart from this gene. Also, *γ-TMT2* (*Glyma.12G014300*) and *γ -TMT1* (*Glyma.12G014200*) are located 703 kb and 711 kb, respectively, apart from the qαTC-12 QTL associated with α-tocopherol content [7]. Similarly, in this study, these candidate genes are located 2.7, 2.6 and 2.6 Mbp apart from *q(γ+ß)-Toc-3-(2017), q(γ+ß)-Toc-4-(2017),* and *qT-Toc-6-(2017),* respectively, on Chr. 12 (Table 4, Appendix A). Also, the TC candidate gene (*Glyma.06G084100*) was located 9.7 Mbp from the QTVEC2_2 QTL underlying the total tocopherol identified earlier [34]. Likewise, this gene is located close to seven seed tocopherol QTL, 2 Mbp from the *q**δ-Toc-3-(2017)* (R^2^ = 27.8)*,* 4.8 and 4.4 Mbp from *q**α-Toc-1-(2017)* and *q**α-Toc-2-(2017)*, respectively, 4.3 and 2.3 Mbp from *qT-Toc-2-(2017)* and *qT-Toc-3-(2017),* respectively, 2.09 Mbp from *q(γ+ß)-Toc-1-(2017),* and 4.5 Mbp from *q**α-Toc-2-(2020)* on Chr. 6 (Table 4, Appendix A). On Chr. 2, the *MPBQ-MT* candidate genes *Glyma.02G002000* and *Glyma.02G143700* are located 1.2 and 4.6 Mbp, respectively from *q**δ-Toc-2-(2017)*. Also, the *Glyma.02G002000* candidate gene is located 0.7 and 4.06 Mbp apart from *q**α-Toc-2-(2020)* and *q(γ+ß)-Toc-1-(2020),* respectively (Table 4, Appendix A). This is coherent with previous studies [34,35], where these two candidate genes were located within seed tocopherol, gamma 1-5 [35] (https://soybase.org/; accessed on 3 April 2022) and seed tocopherol, gamma 2-5 [34] (https://soybase.org/; accessed on 3 April 2022).

The QTL associate with δ-tocopherol explains 27.87% of the phenotype, and the one associated with α-tocopherol explains only 9.96% of the phenotype. A TC (*Glyma.06G084100*) gene was identified close to these QTLs, the TC enzyme is involved directly in the conversion of MPBQ to δ-tocopherol, and indirectly in the conversion to α-tocopherol (Table 4, Figure 2). α-tocopherol is the most known potent fat-soluble antioxidant, it is preferentially absorbed and accumulated in humans [38], its activity has been demonstrated in the prevention and treatment of heart disease, cancer and Alzheimer’s disease [39]. Alpha-tocopherol has been designated as the most beneficial tocopherol compound among health professionals. Unfortunately, this compound is present in small amount in soybean oil when compared to sunflower, canola or corn oil [40]. Therefore, improving α-tocopherol content in soybean is a priority for the soy-industry, the identification in the two years data of the qα-Toc-2-IL-2017 (192.6–197.6 cM) and qα-Toc-3-IL-2020 (195–197 cM) that were collocated on Chr. 06 will provide an opportunity for breeding lines with high α-tocopherol.

Interestingly, the *HPT* candidate gene (*Glyma.17G061900*) is located 4 Mbp apart from qαγR-17 QTL associated with α-tocopherol [7]. Similarly, this candidate gene is located 1.06 Mbp apart from *qT-Toc-5-(2020)* identified here on Chr. 17 (Table 4, Appendix A). The two *HPT* candidate genes *Glyma.10G070100* and *Glyma.10G070300* and the *MPBQ-MT* candidate gene *Glyma.10G030600* are located 3.3, 3.18, and 7.6 Mbp, respectively from the seed tocopherol, total 2-2 QTL [34,35] (https://soybase.org/; accessed on 3 April 2022). Likewise, in this study these genes are located 2.95, 3.04 and 1.28, respectively from *q**δ-Toc-2-(2020)*, and *qT-Toc-3-(2020)* on Chr. 10 (Table 4, Appendix A). The *MPBQ-MT* candidate gene *Glyma.20G211500* and the *HPT* candidate gene (*Glyma.20G245100*) were located 0.22 and 0.9 Mbp apart from seed tocopherol, QTVEC2_2 QTL controlling total tocopherol content identified earlier [34], however, the QTL identified in this study on Chr. 20 was located more than 40 Mbp apart from these genes. Moreover, *Glyma.04G082500*, *Glyma.04G082300*, and *Glyma.04G082400* tocopherol cyclase candidate genes were located within qδTC-4 QTL associated with δ-tocopherol and identified earlier [7].

Although previous studies have reported some soybean genes as candidates for the tocopherol biosynthesis pathway [41], the present study shows the most comprehensive analysis of the whole soybean genome, showing the potential candidate genes for the tocopherol biosynthetic pathway in soybean.

Most QTL regions that were identified in 2017 were not found in 2020 except the qα-Toc-2-IL-2017 (192.6–197.6 cM) and qα-Toc-3-IL-2020 (195–197 cM) that were collocated on Chr. 06. This could be explained by the difference in weather conditions between 2017 and 2020. In August 2017 the temperature ranged between 8 and 33.3 °C, while in August 2020 the temperature ranged between 13.3 and 32.8 °C (https://www.extremeweatherwatch.com/; accessed on 3 April 2022). It has been proven that temperature stress during all stages of development affect soybean seed tocopherol concentrations [42].

The QTL region identified on Chr.7, qTotal-Toc-4-IL-2017 (81.4–88.9 cM), is 7.6 cM from a QTL region previously identified [7]. Likewise, the QTL region identified on Chr. 20 is 61.21 cM away from a QTL region identified in previous studies [34,35]. Interestingly, the QTL region identified on Chr.12 qγ+ß-Toc-3-IL-2017, qγ+ß-Toc-4-IL-2017, qTotal-Toc-6-IL-2017 (0.5–18.5 cM) and Chr.8 qγ+ß-Toc-2-IL-2017 and qT-Toc-2-IL-2020 (12.9–31.2 cM) were reported in previous studies [7,34]. Which make them important regions to investigate further for candidate genes. The rest of the QTLs are novel (Table 4, Appendix A).

Although previous studies identified QTL regions for soybean seed tocopherol content on Chr.6, all the identified QTLs map to the region between 74.5 and 118.5 cM (Appendix A) [7,34,35]. The QTL regions identified in this study on Chr.6 clusters between 173.7 and 207 cM, which is the region that encompass an important gene in the tocopherol biosynthesis pathway, namely the tocopherol cyclase candidate gene, *Glyma.06G084100*. This QTL on Chr.6 is responsible for 27.8% of δ-tocopherol, 9.96% of α-tocopherol, 6.16% of γ+ß-tocopherol, and 6.95% of total tocopherol content

## 4. Materials and Methods

### 4.1. Plant Materials

The F_6:13_ ‘Forrest’ × ‘Williams 82’ RIL population (*n* = 306) described previously was used in this study [30,43]. The parents and RILs were grown in Carbondale, southern Illinois in 2017 and 2020, and seeds were harvested at maturity of all RILs and parents.

### 4.2. Tocopherols Quantification

At maturity, seeds of the parents and RILs were harvested and analyzed for α-Tocopherol (α-Toc), δ-Tocopherol (δ-Toc), α+ß-Tocopherol ((γ+ß)-Toc), and total-Tocopherols (T-Toc) using a protocol developed and validated in the Nguyen Lab, the University of Missouri. Briefly, approx. 1gr. of soybean seeds were ground to fine powders with a Thomas Wiley Mini-Mill, followed by lyophilizing for 48 hrs. Approx. 200mg of powder were mixed with 2mL 200-proof ethanol and vortexed, followed by an incubation with agitation at 75 °C for 2 hrs. The products were then filtered into HPLC vials for analysis along with standard solutions of tocopherols. Quantification of tocopherols was performed by employing an external calibration curve method, in which each curve was created with the six standard solutions of 0.62, 1.25, 2.5, 5, 10, and 20 μg/mL.

### 4.3. DNA Isolation, SNP Genotyping, and Genetic Map Construction

DNA Isolation, SNP Genotyping, and the construction of the F×W82 genetic linkage map have been described earlier [30]. Briefly, SNP genotyping was performed with Illumina Infinium SoySNP6K BeadChips (Illumina, Inc. San Diego, CA, USA) and the genetic map was constructed with JoinMap 4.0 software with a LOD score of 3.0 and maximum distance of 50 cM as described earlier [30].

### 4.4. Seed Tocopherols QTL Detection

We used WinQTL Cartographer 2.5 [31] and both interval mapping (IM) and composite interval mapping (CIM) methods to identify QTL that control seed α-Toc, δ-Toc, γ+ß-Toc, and T-Toc in this RIL population; however, only QTL detected with CIM are reported here. QTL identified via IM are reported in the Appendix A. MapChart 2.2 [32] was used to draw chromosomes with CIM tocopherols QTL locations.

### 4.5. Tocopherols Candidate Genes Identification

The reverse blast of the genes underlying the tocopherol pathway in *Arabidopsis* was conducted using the available data at SoyBase (https://soybase.org/; accessed on 3 April 2022). The sequences of the *Arabidopsis* genes were obtained from the Phytozome database (https://phytozome-next.jgi.doe.gov; accessed on 3 April 2022), these sequences were used for Blast in SoyBase. The obtained genes that control the tocopherol biosynthetic pathway were mapped to the identified tocopherol QTL.

### 4.6. Expression Analysis

The expression analysis of the identified tocopherol candidate genes that are located within or close the identified seed tocopherol QTLs was performed using the publicly available data from SoyBase (https://soybase.org/; accessed on 3 April 2022) to produce the expression profiles of these candidate genes in the soybean reference genome Williams 82 in Glyma1.0 Gene Models version.

## 5. Conclusions

In conclusion, 32 QTL controlling seed tocopherol contents on Chr. 1, 2, 5, 6, 7, 8, 9, 10, 12, 13, 16, 17, and 20 were identified. 37 candidate genes involved in soybean tocopherol biosynthetic pathway have also been identified among which 11 were located close to the QTL regions identified in this study. Two of these candidate genes were highly expressed in seeds *Glyma.05G026200* and *Glyma.17G100700*, followed by *Glyma.06G084100*, *Glyma.02G002000*, *Glyma.02G143700*, *Glyma.10G030600*, and *Glyma.12G014300* with moderate expression profiles in seeds (Figure 3B).

Forrest and Williams 82 sequences of the eleven candidate genes located close to the identified QTLs were compared, and the results have shown that three of them have SNPs between the Forrest and Williams 82 sequences, *Glyma.06G084100, Glyma.17G061900* and *Glyma.17G100700* (Figure 4). The *TC* candidate gene *Glyma.06G084100* has 5 SNPs in the coding sequence, one of them caused a missense mutation (T379A) (Figure 4) in addition to 12 SNPs and 2 InDels in the 5′UTR region (Appendix A). The *HPT* candidate gene, *Glyma.17G061900*, has only one SNP located in the coding sequence that caused a missense mutation (G326A) (Figure 4). For the *GGDR* candidate gene, *Glyma.17G100700*, there is also only one SNP that caused a silent mutation (Figure 4). These SNPs could play a role in the difference of tocopherol content between Forrest and Williams 82 cultivars. *Glyma.06G084100* is associated with the qδ-Toc-3-(2017) (R2 = 27.8), qα-Toc-1-(2017), qα-Toc-2-(2017), qT-Toc-2-(2017), qT-Toc-3-(2017), q(γ+ß)-Toc-1-(2017), and qα-Toc-2-(2020) on Chr. 6 (Table 3, Appendix A). While *Glyma.17G061900* and *Glyma.17G100700* are associated to qT-Toc-5-(2020) on Chr. 17 (Table 2, Appendix A). These genes could be used in breeding programs or gene editing technology to develop soybean lines and cultivars that produce high amounts of the beneficial tocopherols (vitamin E) for human consumption.

## 6. Patents

Patent resulting from this work is under submission.

## Figures and Tables

**Figure 1 plants-11-01258-f001:**
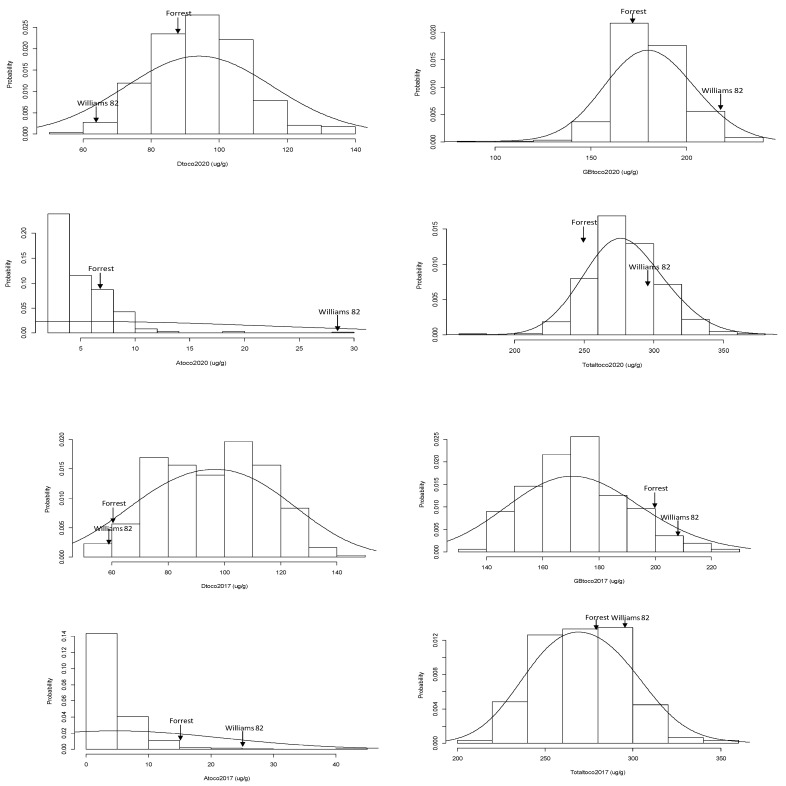
The distribution of seed tocopherol contents (μg/g of seed weight) in the FxW82 RIL population. The seed α-Tocopherol (α-Toc), δ-Tocopherol (δ-Toc), (γ+β)-Tocopherol ((γ+β)-Toc), and Total-Tocopherols (T-Toc) contents were tested in the RILs harvested in Carbondale, IL 2017 and 2020, respectively.

**Figure 2 plants-11-01258-f002:**
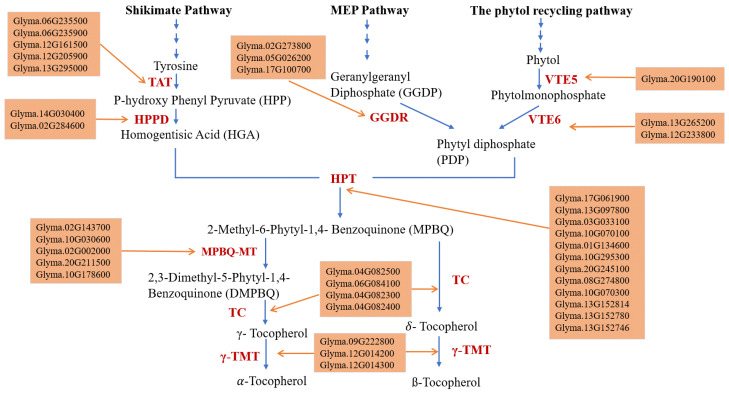
Tocopherol metabolic pathway [15] with identified candidate genes in soybean. TAT: Tyrosine Aminotransferase; HPPD: Hydroxyphenylpyruvate Dioxygenase; GGDR: Geranylgeranyl Diphosphate Reductase; HPT: Homogentisate Phytyl Transferase; MPBQ-MT: methylphytylbenzoquinol methyltransferase; TC: tocopherol cyclase; γ-TMT: gamma tocopherol methyltransferase. MEP Pathway: methylerythritol 4-phosphate pathway.

**Figure 3 plants-11-01258-f003:**
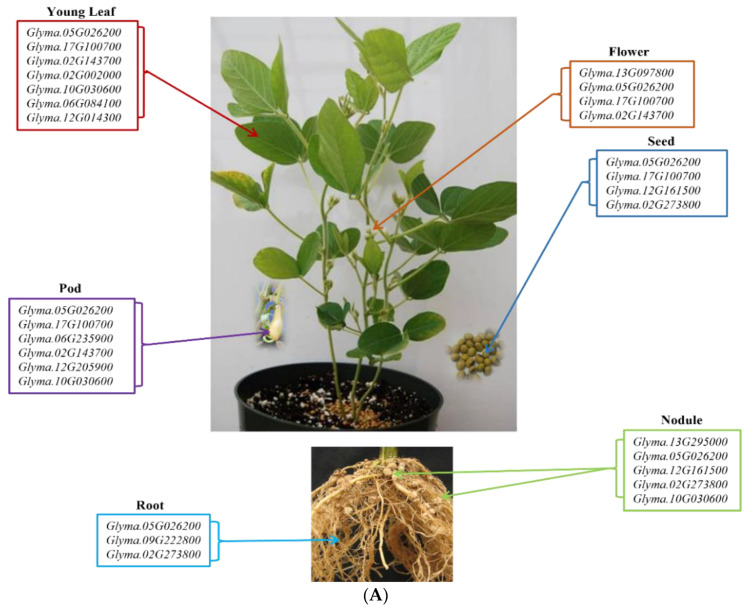
(**A**). Tissue specific expression of the identified tocopherol candidate genes. (**B**). Expression pattern of the 11 tocopherol candidate genes located within tocopherol QTL in Williams 82 (RPKM) were retrieved from publicly available RNA-seq data from Soybase database (http://www.soybase.org/soyseq; accessed on 3 April 2022).

**Figure 4 plants-11-01258-f004:**
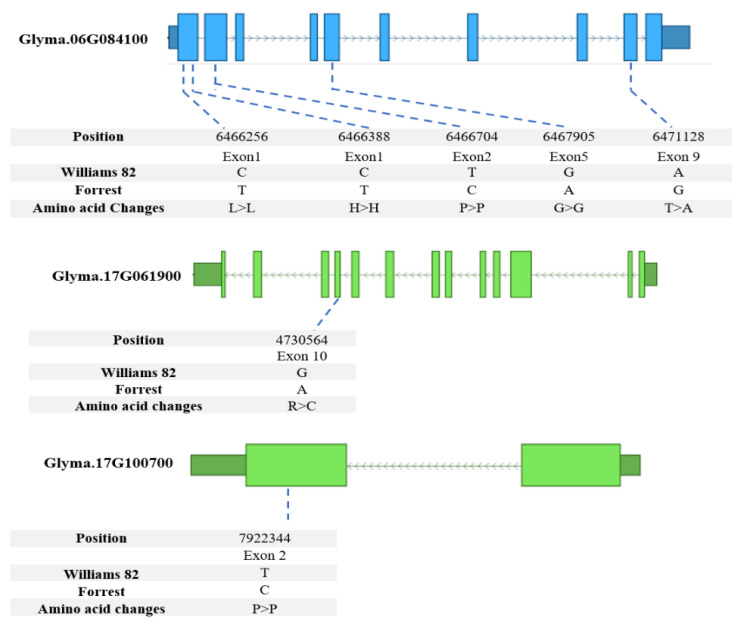
Positions of SNPs between Forrest and Williams 82 cultivars in *Glyma.06G084100, Glyma.17G061900* and *Glyma.17G100700* coding sequences. In the gene model diagram, the light blue/light green boxes represent exons, blue/green bars represent introns, dark blue/dark green boxes represent 3′UTR or 5′UTR. SNPs were positioned relative to the genomic position in the genome version W82.a2.

**Table 1 plants-11-01258-t001:** Mean, range, CV (%), skewness, kurtosis, and value of Shapiro-Wilk normality test (W value) for seed tocopherol content of the RILs in Carbondale, IL. SE: Standard error.

Year	Trait	Mean (μg/g)	Range (μg/g)	CV (%)	SE	Skewness	Kurtosis	W Value (*p* < 0.05)
2017	δ-tocopherol17	95.86	94	19.73	1.09	0.01	2.21	0.98 **
	γ+β-tocopherol17	172.44	97.1	9.78	0.97	0.48	3.18	0.98 **
	α-tocopherol17	4.94	40.7	83.64	0.24	4.46	32.89	0.52 ***
	Total-tocopherol17	271.5	150.7	8.97	1.41	0.25	2.9	0.99
2020	δ-tocopherol20	94.5	76	14.67	0.81	0.2	3.1	0.99
	γ+β-tocopherol20	180.67	149.2	10.06	1.06	−0.36	5.59	0.97 ***
	α-tocopherol20	5.13	25.7	55.49	0.16	3.13	21.28	0.71 ***
	Total-tocopherol20	279.44	206.4	9.13	1.48	0.16	4.32	0.98 **

** *p* < 0.01, *** *p* < 0.001.

**Table 2 plants-11-01258-t002:** The broad sense heritability (h^2^) of tocopherol traits (δ-tocopherol, (γ+β)-tocopherol, α-tocopherol, and total-tocopherol) from the seeds harvested at Carbondale, IL in 2017 and 2020.

**Response: δ-tocopherol**				
	Df	Sum Sq	Mean Seq	H^2^
Line	592	163,429	276.06	0.71
Year	1	391	391.28	
Line:Year	1	81	80.65	
Residuals	0	0	NA	
**Response: γ+β-tocopherol**				
	Df	Sum Sq	Mean Seq	H^2^
Line	592	191,945	324.23	-0.41
Year	1	6	6.13	
Line:Year	1	457	456	
Residuals	0	0	NA	
**Response: α-tocopherol**				
	Df	Sum Sq	Mean Seq	H^2^
Line	592	7479.9	12.635	−0.61
Year	1	20	20.041	
Line:Year	1	20	20.41	
Residuals	0	0	NA	
**Response: Total-tocopherol**				
	Df	Sum Sq	Mean Seq	H^2^
Line	592	377,872	638.3	0.47
Year	1	205	205.28	
Line:Year	1	338	337.55	
Residuals	0	0	NA	

**Table 3 plants-11-01258-t003:** QTLs that control seed α-Tocopherol (α-Toc), δ-Tocopherol (δ-Toc), (α+ß)-Tocopherol ((α+ß)-Toc), and Total-Tocopherols (T-Toc) contents in two environments over two years (A. 2017 and B. 2020). The two environments are in Carbondale, IL (2017) (A) and (2020) (B). Only solid QTL with LOD scores >2.5 and identified by CIM are reported.

**A. QTL that Control Seed Tocopherols Contents in Carbondale, IL (2017)**								
**Trait**	**QTL**	**Chr.**	**Marker/Interval**	**Position (cM)**	**LOD**	**R2**	**Additive**	**Environment**
α-Tocopherol	qα-Toc-1	6	Gm06_1537675-Gm06_1570293	173.7–178.7	6.1	9.96	1.648195	Carbondale, IL
	qα-Toc-2	6	Gm06_1858327-Gm06_2048675	192.6–197.6	6.77	9.95	1.477826	Carbondale, IL
δ-Tocopherol	qδ-Toc-1	1	Gm01_1887205-Gm01_1653315	174.2–179.2	3.06	3.1	−3.30536	Carbondale, IL
	qδ-Toc-2	2	Gm02_1481798-Gm02_9925870	133.5–140.2	3.4	3.57	5.481172	Carbondale, IL
	qδ-Toc-3	6	Gm06_1674534-Gm06_4447485	183.8–207	23.01	27.9	10.14229	Carbondale, IL
γ+ß-Tocopherol	qγ+ß-Toc-1	6	Gm06_1674534-Gm06_4368839	185.8–203.2	5.13	6.16	4.161084	Carbondale, IL
	qγ+ß-Toc-2	8	Gm08_3018731-Gm08_4266625	17.8–31.2	3.02	3.78	−3.23578	Carbondale, IL
	qγ+ß-Toc-3	12	Gm12_3820261-Gm12_3818392	0.5–1	4.14	5.23	3.852091	Carbondale, IL
	qγ+ß-Toc-4	12	Gm12_3805393-Gm12_3696093	2.5–18.5	7.18	8.86	5.050705	Carbondale, IL
	qγ+ß-Toc-5	13	Gm13_2587196-Gm13_2048499	189.1–210.7	3.79	5.43	3.906039	Carbondale, IL
Total- Tocopherols	qTotal-Toc-1	5	Gm05_3674925-Gm05_3256515	29.4–32.2	3.63	4.42	8.695197	Carbondale, IL
	qTotal-Toc-2	6	Gm06_1739930-Gm06_2073990	188.9–197.6	4.07	5.05	−5.69939	Carbondale, IL
	qTotal-Toc-3	6	Gm06_3849946-Gm06_4447485	200.1–207	5.67	6.95	−6.57105	Carbondale, IL
	qTotal-Toc-4	7	Gm07_3635708-Gm07_1829304	81.4–88.9	2.88	3.46	−5.31938	Carbondale, IL
	qTotal-Toc-5	9	Gm09_3483063-Gm09_3544488	74.8–78	3.13	3.78	−4.56753	Carbondale, IL
	qTotal-Toc-6	12	Gm12_3820261-Gm12_3696093	2.5–18.5	4.84	5.94	4.753021	Carbondale, IL
**B. QTLs that control seed tocopherols contents in Carbondale, IL (2020)**								
**Trait**	**QTL**	**Chr.**	**Marker**	**Position (cM)**	**LOD**	**R2**	**Additive**	**Envt.**
α-Tocopherol	qα-Toc-1	1	Gm01_3466825-Gm01_5255151	4.1–10.1	5.81	0.35	2.35	Carbondale, IL
	qα-Toc-2	2	Gm02_5141136-Gm02_1020061	137.1–139.8	2.9	0.04	0.82	Carbondale, IL
	qα-Toc-3	6	Gm06_1954068-Gm06_2015292	195–197	2.01	0.03	0.5	Carbondale, IL
δ-Tocopherol	qδ-Toc-1	1	Gm01_4912170-Gm01_4852475	91.6–93.6	2.57	0.04	−2.66	Carbondale, IL
	qδ-Toc-2	8	Gm08_1810148-Gm08_2201336	125.7–130.8	2.42	0.04	2.86	Carbondale, IL
	qδ-Toc-3	10	Gm10_3943637-Gm10_3935014	79.2–81.4	2.4	0.03	−2.53	Carbondale, IL
	qδ-Toc-4	16	Gm16_1079308-Gm16_3673245	0.5–12.5	3.71	0.05	−6.65	Carbondale, IL
	qδ-Toc-5	20	Gm20_3665142-Gm20_1046460	174.9–176.9	2.68	0.04	−5.95	Carbondale, IL
(γ+ß)-Tocopherol	q(γ+ß)-Toc-2	2	Gm02_5155733-Gm02_4311734	130.5–132.5	2.15	0.04	−9.75	Carbondale, IL
	q(γ+ß)-Toc-1	16	Gm16_1079308-Gm16_3673245	2.5–18.5	2.87	0.23	−10.56	Carbondale, IL
Total- Toopherol	qT-Toc-1	1	Gm01_3504836-Gm01_5566630	0.1–1.7	4.52	0.08	−14.62	Carbondale, IL
	qT-Toc-2	8	Gm08_2622664-Gm08_2852874	12.9–13.3	2.11	0.03	−4.44	Carbondale, IL
	qT-Toc-3	10	Gm10_3935014-Gm10_3890052	79.4–84.4	2	0.03	−4.27	Carbondale, IL
	qT-Toc-4	16	Gm16_1079308-Gm16_3673245	0.5–18.5	3.1	0.18	−13.92	Carbondale, IL
	qT-Toc-5	17	Gm17_3916734-Gm17_3929518	6.2–48.7	3.07	0.18	−11.81	Carbondale, IL
	qT-Toc-6	20	Gm20_3665142-Gm20_1046460	174.9–176.9	2.51	0.03	−10.65	Carbondale, IL

**Table 4 plants-11-01258-t004:** Tocopherol candidate genes located within or close to the tocopherol QTL identified in the FxW82 RIL population grown in Carbondale, IL over two years **A.** 2017 and **B.** 2020.

**A. QTLs that Control Seed Tocopherols Contents in Carbondale, IL (2017)**								
**Trait**	**QTL**	**Wm82.a4. v1 Gene Models**			**Glyma1.0 Gene Models**			
		**Gene ID**	**Start**	**End**	**Gene ID**	**Start**	**End**	**Dist. (Mbp)**
α-Tocopherol	qα-Toc-1	*Glyma.06G084100*	6,435,516	6,441,328	*Glyma06g08850*	6,460,802	6,466,636	4.8
	qα-Toc-2	*Glyma.06G084100*	6,435,516	6,441,328	*Glyma06g08850*	6,460,802	6,466,636	4.4
δ-Tocopherol	qδ-Toc-1							
	qδ-Toc-2	*Glyma.02G002000*	237,750	243,006	*Glyma02g00440*	237,612	245,017	1.2
		*Glyma.02G143700*	15,253,811	15,256,708	*Glyma02g16210*	14,623,815	14,626,862	4.6
	qδ-Toc-3	*Glyma.06G084100*	6,435,516	6,441,328	*Glyma06g08850*	6,460,802	6,466,636	2.01
(γ+ß)-Tocopherol	q(γ+ß)-Toc-1	*Glyma.06G084100*	6,435,516	6,441,328	*Glyma06g08850*	6,460,802	6,466,636	2.09
	q(γ+ß)-Toc-2							
	q(γ+ß)-Toc-3	*Glyma.12G014200*	1,020,484	1,023,995	*Glyma12g01680*	1,020,554	1,024,132	2.7
		*Glyma.12G014300*	1,028,051	1,031,954	*Glyma12g01690*	1,028,132	1,032,092	2.7
	q(γ+ß)-Toc-4	*Glyma.12G014200*	1,020,484	1,023,995	*Glyma12g01680*	1,020,554	1,024,132	2.6
		*Glyma.12G014300*	1,028,051	1,031,954	*Glyma12g01690*	1,028,132	1,032,092	2.6
	q(γ+ß)-Toc-5							
Total-Tocopherols	qT-Toc-1	*Glyma.05G026200*	2,284,067	2,286,242	*Glyma05g01000*	606,481	608,812	2.6
	qT-Toc-2	*Glyma.06G084100*	6,435,516	6,441,328	*Glyma06g08850*	6,460,802	6,466,636	4.3
	qT-Toc-3	*Glyma.06G084100*	6,435,516	6,441,328	*Glyma06g08850*	6,460,802	6,466,636	2.3
	qT-Toc-4							
	qT-Toc-5							
	qT-Toc-6	*Glyma.12G014200*	1,020,484	1,023,995	*Glyma12g01680*	1,020,554	1,024,132	2.6
		*Glyma.12G014300*	1,028,051	1,031,954	*Glyma12g01690*	1,028,132	1,032,092	2.6
**B. QTLs that control seed tocopherols contents in Carbondale, IL (2020)**								
**Trait**	**QTL**	**Wm82.a4. v1 Gene Models**			**Glyma1.0 Gene Models**			
		**Gene ID**	**Start**	**End**	**Gene ID**	**Start**	**End**	**Dist. (Mbp)**
α-Tocopherol	*q* *α-Toc-1*							
	*q* *α-Toc-2*	*Glyma.02G002000*	237,689	243,112	*Glyma02g00440*	237,612	245,017	0.7
	*q* *α-Toc-3*	*Glyma.06G084100*	6,466,090	6,471,839	*Glyma06g08850*	6,460,802	6,466,636	4.5
δ-Tocopherol	*q* *δ-Toc-1*							
	*q* *δ-Toc-2*							
	*q* *δ-Toc-3*	*Glyma.10G030600*	2,658,064	2,661,302	*Glyma10g03590*	2,650,012	2,653,309	1.28
		*Glyma.10G070100*	6,923,409	6,931,780	*Glyma10g08080*	6,888,551	6,893,731	2.95
		*Glyma.10G070300*	7,023,173	7,029,710	*Glyma10g08150*	6,986,426	6,992,505	3.04
	*q* *δ-Toc-4*							
	*q* *δ-Toc-5*							
(γ+ß)-Tocopherol	*q(γ+ß)-Toc-2*	*Glyma.02G002000*	237,689	243,112	*Glyma02g00440*	237,612	245,017	4.06
	*q(γ+ß)-Toc-1*							
Total-Tocopherol	*qT-Toc-1*							
	*qT-Toc-2*							
	*qT-Toc-3*	*Glyma.10G030600*	2,658,064	2,661,302	*Glyma10g03590*	2,650,012	2,653,309	1.23
		*Glyma.10G070100*	6,923,409	6,931,780	*Glyma10g08080*	6,888,551	6,893,731	2.95
		*Glyma.10G070300*	7,023,173	7,029,710	*Glyma10g08150*	6,986,426	6,992,505	3.05
	*qT-Toc-4*							
	*qT-Toc-5*	*Glyma.17G061900*	4,728,685	4,734,790	*Glyma17g06940*	4,998,801	5,004,742	1.06
		*Glyma.17G100700*	7,920,291	7,923,450	*Glyma17g10890*	8,190,830	8,194,219	4.2
	*qT-Toc-6*							

**Table 5 plants-11-01258-t005:** Tocopherol candidate genes associated to the previously reported QTLs.

Gene ID	Start	End	QTL	QTL Start	QTL End	Parents	Number Loci Tested	Lod Score	Interval Length	Reference
*Glyma.09G222800*	44,341,974	44,346,311	qαTC-9	43,927,286	44,366,371	TK780 X B04009	ND	13.1	ND	[7]
*Glyma.12G014200*	1,026,615	1,029,095	qαTC-12	1,507,927	1,790,872	TK780 X B04009	ND	7.8	ND	[7]
*Glyma.12G014300*	1,033,151	1,037,054	qαTC-12	1,507,927	1,790,872	TK780 X B04009	ND	7.8	ND	[7]
*Glyma.04G082500*	6,948,445	6,954,177	qδTC-4	6,780,105	7,188,146	TK780 X B04009	ND	5.5	ND	[7]
*Glyma.04G082400*	6,946,447	6,947,480	qδTC-4	6,780,105	7,188,146	TK780 X B04009	ND	5.5	ND	[7]
*Glyma.04G082300*	6,945,685	6,946,469	qδTC-4	6,780,105	7,188,146	TK780 X B04009	ND	5.5	ND	[7]
*Glyma.06G084100*	6,466,090	6,471,839	Seed tocopherol, alpha 1-2	16,106,296	16,256,544	OAC Bayfield X Hefeng 25	107	ND	ND	[35]
*Glyma.14G030400*	2,204,142	2,206,424	Seed tocopherol, alpha 2-1	675,214	2204,996	Hefeng 25 X OAC Bayfield	606	ND	ND	[34]
*Glyma.02G143700*	14,826,295	14,829,286	Seed tocopherol, gamma 1-5	13,316,369	37,285,448	OAC Bayfield X Hefeng 25	107	ND	ND	[35]
			Seed tocopherol, gamma 2-5	14,288,241	45,267,040	Hefeng 25 X OAC Bayfield	606	ND	56.73	[34]
*Glyma.02G002000*	237,689	243,112	Seed tocopherol, gamma 1-5	13,316,369	37,285,448	OAC Bayfield X Hefeng 25	107	ND	ND	[35]
			Seed tocopherol, gamma 2-5	14,288,241	45,267,040	Hefeng 25 X OAC Bayfield	606	ND	56.73	[34]
*Glyma.13G097800*	21,299,008	21,305,797	Seed tocopherol, delta 1-3	15,248,933	15,306,234	OAC Bayfield X Hefeng 25	107	ND	ND	[35]
*Glyma.17G061900*	4,728,685	4,734,790	qαγR-17	8,786,113	9,025,866	TK780 X B04009	ND	4.1	ND	[7]
*Glyma.17G100700*	792,0291	7,923,450	Seed tocopherol, gamma 3-6	5,891,979	36,718,722	OAC Bayfield X OAC Shire	550	2.6	67.66	[13]
*Glyma.12g161500*	30,805,424	30,815,155	Seed tocopherol, total 3-5	24,129,662	37,556,592	OAC Bayfield X OAC Shire	550	3.4	29.62	[13]
*Glyma.12g205900*	38,082,220	38,086,113	Seed tocopherol, alpha 3-3	24,129,662	37,556,592	OAC Bayfield X OAC Shire	550	3.5	29.62	[13]
			Seed tocopherol, total 3-5	24,129,662	37,556,592	OAC Bayfield X OAC Shire	550	3.4	29.62	[13]
*Glyma.13g295000*	38,800,738	38,805,839	Seed tocopherol, delta 3-2	37,603,911	40,131,770	OAC Bayfield X OAC Shire	550	2.6	17.11	[13]
			Seed tocopherol, delta 3-3	31,449,060	43,325,731	OAC Bayfield X OAC Shire	550	3.8	52.94	[13]

## Data Availability

Data supporting reported results are available on request from the corresponding author.

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
