# Peer review of "QTL and Candidate Genes for Seed Tocopherol Content in ‘Forrest’ by ‘Williams 82’ Recombinant Inbred Line (RIL) Population of Soybean"

_plants, 2022, doi:10.3390/plants11091258_

Round 1

Reviewer 1 Report

The publication is written clearly, there is no doubt in the description of the results or the methodology.

However, there are a few shortcomings and errors such as gene names written without italics etc. In general, gene symbols should be italicized (eg. IGF1) while protein symbols shouldn’t (eg. IGF1). Additionally, there is no italic in the names of the plants for example: line 52 “Glycine max Merr.” should be “Glycine max L. Merrill”; lines 92, 95 “A. thaliana” etc. In the reference section there is mistake in point 13.

Regarding technical notes: “Table 1” is in low resolution, please complete this. I would also like to ask to add QTL likelihood curves of LOD score as well as the threshold line to the Figure 2 and the tissue names in the Figure 4A.

I have question about the mapping population used in this studies. Which generation of the population of RIL "Forrest" × "Williams 82" was used? There is no information about it in the text.

Unfortunately, there is no supplementary file available, so please complete it.

Author Response

The publication is written clearly, there is no doubt in the description of the results or the methodology.

  • However, there are a few shortcomings and errors such as gene names written without italics etc. In general, gene symbols should be italicized (eg. IGF1) while protein symbols shouldn’t (eg. IGF1). Additionally, there is no italic in the names of the plants for example: line 52 “Glycine max Merr.” should be “Glycine max  Merrill”; lines 92, 95 “A. thaliana” etc. In the reference section there is mistake in point 13.

Answer:

We would like to thank you for the positive review. The names of the genes and the plants were italicized in the original draft, however during the conversion to the MDPI Plant format, it was all lost. We made sure to get it corrected in the current draft.

  • Regarding technical notes: “Table 1” is in low resolution, please complete this. I would also like to ask to add QTL likelihood curves of LOD score as well as the threshold line to the Figure 2 and the tissue names in the Figure 4A.

Answer:

The resolution of Table 1 has been corrected.

The tissue names have been added to the Figure 4A.

Figure S2. QTL that control seed ?-Tocopherol (?-Toc), ?-Tocopherol (?-Toc), (γ+β)-Tocopherol ((γ+β)-Toc), and Total-Tocopherols (T-Toc) contents identified by IM and CIM methods in the F×W82 RIL population grown in two environments in Carbondale, IL over two years (2017 and 2020) was added to the supplementary data.

  • I have question about the mapping population used in this studies. Which generation of the population of RIL "Forrest" × "Williams 82" was used? There is no information about it in the text.

Answer:

It’s a F6:13 generation of the RIL population. This information was added to M&M.

  • Unfortunately, there is no supplementary file available, so please complete it.

Answer:

Supplementary file has been added.

Reviewer 2 Report

In this paper, the QTL loci related to tocopherol are described in a large part, but the emphasis on α-tocopherol is too weak due to the lack of data support.

In the discussion, some sentences are repeated in the research background, which is too deliberate. Some sentences are related to each other and can be combined easily.

There are some problems such as unclear pixel and imperfect format in the full text table chart, which should be improved.

Author Response

  • In this paper, the QTL loci related to tocopherol are described in a large part, but the emphasis on α-tocopherol is too weak due to the lack of data support.

Answer:

A paragraph about the alpha tocopherol was added to the discussion as suggested by the reviewer

  • In the discussion, some sentences are repeated in the research background, which is too deliberate. Some sentences are related to each other and can be combined easily.

Answer:

The repeated sentences have been removed.

  • There are some problems such as unclear pixel and imperfect format in the full text table chart, which should be improved.

Answer:

The format problems have been fixed.

Reviewer 3 Report

In the reviewed manuscript entitled 'QTL and Candidate Genes for Seed Tocopherol Content in' Forrest 'By' Williams 82 'Recombinant Inbred Line (RIL) Population of Soybean' a total of 32 QTL controlling various seed tocopherol contents have been identified and mapped on Chrs. 1, 2, 5, 6, 7, 8, 9, 10, 12, 13, 16, 17, and 20. One major and novel QTL was identified on Chr. 6. Reverse Blast analysis allowed the identification of 37 genes involved in soybean tocopherol pathway, among which 11 were located close to the identified QTLs.

The authors did not include Supplementary Files, hence it was impossible to verify all the attached results.

What worries me the most is the lack of QTLs that would be identified in both years of research. The explanation regarding the temperature, which did not differ significantly in the two years, does not seem convincing and the best solution would be to repeat the field experiment in the next year or even years. In my opinion, on the basis of the results obtained, it is difficult to select QTLe that would affect the obtaining of breeding progress.

Many new QTLs have been identified, are you sure they are reliable? In addition, candidate genes have also been identified at great distances from potential loci. It seems that one more field experiment and RNA-seq analysis aimed at identifying genes related to tocopherol synthesis are missing.

The map with marked QTLs seems unnecessary. Should be moved to Supplementary Materials.

There is no need for detailed descriptions of all QTLs. Perhaps it would be worth limiting these descriptions. After all, all the information is in the tables.

The citations are not valid for the journal Plants.

Remember to write Latin names in italics.

Some additional comments are in the attached file.

In my opinion, the work should be rewritten and shortened and limited to describing only the most valuable and promising QTLs.

Author Response

In the reviewed manuscript entitled 'QTL and Candidate Genes for Seed Tocopherol Content in' Forrest 'By' Williams 82 'Recombinant Inbred Line (RIL) Population of Soybean' a total of 32 QTL controlling various seed tocopherol contents have been identified and mapped on Chrs. 1, 2, 5, 6, 7, 8, 9, 10, 12, 13, 16, 17, and 20. One major and novel QTL was identified on Chr. 6. Reverse Blast analysis allowed the identification of 37 genes involved in soybean tocopherol pathway, among which 11 were located close to the identified QTLs.

  • The authors did not include Supplementary Files; hence it was impossible to verify all the attached results.

Answer:

We would like to thank you for your valuable reviews. The supplementary files have been added.

  • What worries me the most is the lack of QTLs that would be identified in both years of research. The explanation regarding the temperature, which did not differ significantly in the two years, does not seem convincing and the best solution would be to repeat the field experiment in the next year or even years. In my opinion, on the basis of the results obtained, it is difficult to select QTLe that would affect the obtaining of breeding progress.Many new QTLs have been identified, are you sure they are reliable? In addition, candidate genes have also been identified at great distances from potential loci. It seems that one more field experiment and RNA-seq analysis aimed at identifying genes related to tocopherol synthesis are missing.

Answer:

One of the major purposes of this manuscript is to identify the genes linked to the tocopherol biosynthesis pathway in soybean via mapping and/or the information from other plants including the model plant Arabidopsis, using the information from Arabidopsis, we were able to identify the candidate genes involved in the tocopherol pathway in soybean, and link some of them to the identified QTLs in this study and to the previously reported ones.

I agree with the statement made by the reviewer, if we have the opportunity to repeat the study, we will do that to confirm additional QTLs for tocopherol, however those compounds are highly linked to the environmental impact, minor fluctuations in temperature could have a huge impact on the phenotype, and it will be hard to identify the majority of the QTLs in two years of data, as it has been also a challenge for most investigators in the field. Among the leading labs in the field, Park et al., 2019, identified 22 QTLs controlling tocopherol content in soybean seeds, among them only 4 QTLs were identified in both years.

In our study, from the 32 QTLs that we identified, we were able to confirm 2 in our two years data, however we were able to confirm five QTLs from our study to the same chromosomal locations when compared to previous study (Park et al.,2019; and Li et al., 2016), and two other QTLs to the same chromosomes (Park et al.,2019; Li et al., 2016; and Li et al., 2010)

  • The map with marked QTLs seems unnecessary. Should be moved to Supplementary Materials.

Answer:

The figure 2 has been moved to the Supplementary Materials (Figure S1).

  • There is no need for detailed descriptions of all QTLs. Perhaps it would be worth limiting these descriptions. After all, all the information is in the tables.

Answer:

Most of the paragraphs that were duplication of what was presented in the tables and figures were removed from the text as suggested. A focus has been made on the alpha tocopherol QTL that was confirmed in two years data.

  • The citations are not valid for the journal Plants.

Answer:

The citations have been corrected.

  • Remember to write Latin names in italics.

Answer:

The names of the genes and the plants were italicized in the original draft, however during the conversion to the MDPI Plant format, it was all lost. We made sure to get it corrected in the current draft.

  • Some additional comments are in the attached file.

Answer:

The manuscript was edited following the comments. We would like to thank the reviewer.

  • In my opinion, the work should be rewritten and shortened and limited to describing only the most valuable and promising QTLs.

Answer:

Most of the paragraphs that were duplication of what was presented in the tables and figures were removed from the text as suggested. A focus has been made on the alpha tocopherol QTL that was confirmed in two years data.

Round 2

Reviewer 2 Report

The major problems in the paper have been solved, and the author needs to further check the paper to avoid minor mistakes.

Reviewer 3 Report

The current version of the manuscript is better than the previous one acceptable for publishing.